∂ | **Open Peer Review** | Host-Microbial Interactions | Research Article

# Preliminary evidence for chaotic signatures in host-microbe interactions

Yehonatan Sella,[1] Nichole A. Broderick,[2] Kaitlin M. Stouffer,[3] Deborah L. McEwan,[4] Frederick M. Ausubel,[4] Arturo Casadevall,[3] Aviv Bergman[1,5]

**ABSTRACT**  Host-microbe interactions constitute dynamical systems that can be represented by mathematical formulations that determine their dynamic nature and are categorized as deterministic, stochastic, or chaotic. Knowing the type of dynamical interaction is essential for understanding the system under study. Very little experimental work has been done to determine the dynamical characteristics of host-microbe interactions, and its study poses significant challenges. The most straightforward experimental outcome involves an observation of time to death upon infection. However, in measuring this outcome, the internal parameters and the dynamics of each particular host-microbe interaction in a population of interactions are hidden from the experimentalist. To investigate whether a time-to-death (time-to-event) data set provides adequate information for searching for chaotic signatures, we first determined our ability to detect chaos in simulated data sets of time-to-event measurements and successfully distinguished the time-to-event distribution of a chaotic process from a comparable stochastic one. To do so, we introduced an inversion measure to test for a chaotic signature in time-to-event distributions. Next, we searched for chaos in the time-to-death of *Caenorhabditis elegans* and *Drosophila melanogaster* infected with *Pseudomonas aeruginosa* or *Pseudomonas entomophila*, respectively. We found suggestions of chaotic signatures in both systems but caution that our results are preliminary and highlight the need for more fine-grained and larger data sets in determining dynamical characteristics. If validated, chaos in host-microbe interactions would have important implications for the occurrence and outcome of infectious diseases, the reproducibility of experiments in the field of microbial pathogenesis, and the prediction of microbial threats.

**IMPORTANCE**  Is microbial pathogenesis a predictable scientific field? At a time when we are dealing with coronavirus disease 2019, there is intense interest in knowing about the epidemic potential of other microbial threats and new emerging infectious diseases. To know whether microbial pathogenesis will ever be a predictable scientific field requires knowing whether a host-microbe interaction follows deterministic, stochastic, or chaotic dynamics. If randomness and chaos are absent from virulence, there is hope for prediction in the future regarding the outcome of microbe-host interactions. Chaotic systems are inherently unpredictable, although it is possible to generate short-term probabilistic models, as is done in applications of stochastic processes and machine learning to weather forecasting. Information on the dynamics of a system is also essential for understanding the reproducibility of experiments, a topic of great concern in the biological sciences. Our study finds preliminary evidence for chaotic dynamics in infectious diseases.

**KEYWORDS**  chaos, host-parasite relationship, dynamical system

Address correspondence to Arturo Casadevall, acasade1@jhu.edu, or Aviv Bergman, aviv@einsteinmed.edu.

Yehonatan Sella and Nichole A. Broderick contributed equally to this article. Author order was determined based on the area of contribution to this paper. Since this is primarily a mathematical analysis of a biological phenomenon done by Y.S. and A.B., these authors were placed in the first and last positions, respectively.

Arturo Casadevall and Aviv Bergman contributed equally to this article.

The authors declare no conflict of interest.

Host-microbe interactions can have variable outcomes that result in at least four states, namely mutualism, commensalism, latency, and disease (1, 2). These states can be distinguished by the amount of damage incurred on the host. When damage is sufficient to affect homeostasis, disease ensues and is manifested as symptoms of distress. When damage is sufficient to cripple essential systems, death ensues. Studying the dynamics of host-microbe interactions is inherently difficult, since measurements of host-microbe interactions, such as survival time post-infection, are often very difficult to reproduce quantitatively even when the initial conditions appear to be nearly identical. There are several non-exclusionary explanations for the lack of detailed inter-experimental reproducibility even when the overall result is reproducible. On one hand, microbial pathogenesis is dependent on a dizzying array of variables that include microbe- and host-specific factors, such as inoculum size, route of infection, host state, temperature, associated microbiota, etc. (3) Even in the most carefully conducted experiment, it is unlikely that all variables are perfectly controlled from experiment to experiment. This is apparent even in experiments where a qualitative variable such as host death is consistent but quantitative outcomes such as time to death can vary significantly from experiment to experiment (4). For instance, even within an experiment where a set of genetically identical hosts in the form of inbred mice are infected with the same inoculum of a bacterium or parasite, organ microbial burden across individuals can vary by 100-fold (for an example, see Fig. 2 in reference 5). Such ranges are often attributed to experimental variation, although to our knowledge the sources of such variability have not been exhaustively investigated in any host-microbe interaction, and such studies may not be feasible with current experimental capabilities. This interpretation assumes that the system is deterministic since it operates on the implicit assumption that if an experimenter could control for all experimental variables, the results would be perfectly reproducible from experiment to experiment. Yet, this assumption has never been formally tested, and it is possible that experimental noise and lack of reproducibility in host-microbe interactions also reflect the mathematical properties of the systems under study.

Dynamical systems are either stochastic or deterministic. Stochastic systems are dominated by random variables and are inherently unpredictable. Deterministic systems are predictable in that they produce the same output given the same initial conditions. Chaotic systems are a subset of deterministic systems where the long-term course of the system is highly unpredictable, even while the system is theoretically predictable. This seeming contradiction stems from the high sensitivity to initial conditions possessed by a chaotic system, popularly captured by the notion of "the butterfly effect," where a small change to the weather system such as a butterfly flapping its wings cascades onto a large change in the system later, potentially culminating in a tornado. In a deterministic system, the future course of the system is theoretically possible to determine, if given perfect knowledge of the current state of the system and its dynamics. But since in practice, we can never account fully and with complete precision for all the state variables, such long-term prediction is impossible. By the same token, experiments carried out in the context of a chaotic system cannot be expected to be repeatable since we cannot control for all variables with perfect precision. Here, we carried out an exploration of the dynamics of host-microbe interactions by first posing a thought experiment, or thought hypothesis, where we consider whether an experiment would be perfectly reproducible if all the variables could be controlled. Since chaotic systems are deterministic, such an experiment would, theoretically, in principle, be reproducible, but in reality, small variations in repeating the experiment are inevitable, and consequently, final outcomes would differ. We emphasize to the non-specialist reader that mathematical chaos does not mean "chaotic" in vernacular parlance. The free online dictionary defines chaos as "a condition or place of great disorder or confusion," but the mathematical definition is that of a system that is highly sensitive to the initial conditions, which is quantified by the maximal Lyapunov exponent being greater than zero, indicating that nearby trajectories at a given point in time diverge exponentially with respect to time in

the future. Chaotic systems are to be contrasted with stochastic, or random, dynamical systems. In stochastic systems, neither short- nor long-term prediction is possible due to an irreducible element of randomness. For example, the outcome of a gambling game where the cards are dealt at random cannot be predicted because complete chance determines the combinations at various points. Importantly, it has been observed that in biology, chaotic behavior is prevalent; however, as noted by Rogers et al. (6), it is often somewhat difficult to conclusively establish a chaotic signature due to a lack of sufficient data, as is the case in the current study.

There is very little information available on the dynamical characteristics of host-microbe interactions, i.e., whether these dynamics are stochastic, predictably deterministic, or chaotic. *Cryptococcus neoformans* infection in the moth *Galleria mellonella* followed a deterministic and predictable course, and no signatures of chaos were detected, but the inoculum used may have been too large, thereby resulting in a predictable outcome of death that overwhelmed any chaotic influences in this system (7). In contrast, analysis of influenza virus dynamics (8) and host-parasitoid systems (9) suggests dynamical behavior suggestive of chaos. Chaos is believed to be widespread in the biological world and has been reported either empirically or theoretically in predator-prey relationships (10), microbial densities in ecological systems (10), DNA content during mitosis (11), and even in processes that are considered periodic such as cardiac rhythms (12), circadian rhythms (13), gene regulation (14), human menstrual cycles (15), signaling cascades (16), and yeast budding cycles (17). Given the likelihood of the existence of chaotic signatures in such diverse systems, it is worthwhile to consider whether it also occurs in microbial pathogenesis, a process that involves emergent properties arising from the host-microbe interaction. In this study, we investigated the dynamics of *Pseudomonas* spp. infection in *Caenorhabditis elegans* and *Drosophila melanogaster*.

## RESULTS

### Detecting chaos in time-to-death distributions

When faced with a time series sampled from some dynamical process, one might want to detect whether the underlying process is chaotic or stochastic. Of course, there could be some stochasticity on top of an underlying chaotic process, in which case one might still want to detect the presence of chaos. The difficulty arises from the fact that many statistics of a chaotic process resemble a stochastic one—the determinism of the process rendered barely detectable due to the unstable dynamical trajectories characteristic of chaos. However, methods exist for distinguishing chaos (18, 19). One such method is the permutation spectrum test, which studies the distribution of ordinal patterns in short consecutive subsequences of the time series (20, 21). In particular, the existence of forbidden ordinal patterns—those with no or almost no occurrence—is known as a strong indication of chaos, which distinguishes it from a truly stochastic time series. In the experiments described here, a population of hosts is exposed to pathogens, and the resulting time-to-death distribution is compared to that of a naturally occurring time-to-death distribution of a control population. In this case, the problem of detecting underlying chaos in the dynamics of host-microbe interactions, when the readout is time-to-death, is complicated by two factors. First, we do not have access to the internal dynamics of the system, only to the time at which a terminal event (death) occurs. Second, since we only gather one data point from each individual (time until death), we must study an assemblage of systems rather than a single continual system. This forces a degree of randomness on the data, especially as we cannot control the system's internal parameters. In particular, we can no longer use the permutation spectrum test in the traditional way since we have a distribution of time-to-deaths pulled from a sample population rather than a sample of time series from single individual subjects. In addition, we note that in both worms and flies, the infection may not begin at the same time for all animals, as they may ingest the inoculum at different times post-exposure. As such, the time zero reflects the beginning of the experiment rather than the time of initial infection. Nevertheless, as stated in the methods, for both systems, infection is

expected to occur shortly after the experiments begin. Given that this is a physiological infection method where the animals are infected by ingesting the pathogen, repeated infections are also theoretically likely, but neither are common since infected worms remain infected after first injecting bacteria and flies incur such gut damage from the initial infection that they are unlikely to be candidates for subsequent infection. Hence, our dynamic analysis is for the whole infection system rather than for individual animals, and we note that this approach is more akin to what happens in natural epidemics resulting from the encounter of a pathogen with a susceptible host population.

Faced with these problems, we devised a new test that we term an "inversion measure," which we suggest can be used in detecting chaos based only on the distribution of time-to-event in an ensemble system. This method is based on a histogram representation of the time-to-event distribution. Similar to the permutation spectrum test, we subdivide the histogram's bins into consecutive non-overlapping sequences of $k = 4$ bins. For each such sequence, we check whether it is an inversion, meaning the direction of change from second to third bin is opposite to the direction of change from first to fourth. We measure the frequency of four-bin sequences that are inversions. We base this measure on the understanding that distributions that are roughly smooth after averaging away random noise would have fewer inversions, whereas distributions that are less smooth and more irregular, which is a characteristic of chaos (18), would potentially have more inversions. Indeed, as the direction of change from the first to fourth bin is expected to indicate the local linear trend, the direction of change from the second to third bin would be expected, on average, to align with this trend as well. Our measure is inspired by the permutation spectrum test in studying the aggregate behavior of short sequences of time. We assign a $P$-value to the measure based on bootstrapping against a null hypothesis that the distribution is locally linear (see Materials and Methods for more details). Thus, a low $P$-value indicates the distribution significantly deviates from a locally linear one and may have chaotic characteristics. We note, however, that this is not a definitive indicator of chaos but rather of persistent nonlinear irregularities that could arise from chaos. In this paper, we propose this measure as a preliminary test for discriminating between truly stochastic processes and chaotic ones, based on time-to-event distributions. This is a nonparametric test as the null hypothesis makes no assumptions about the specific shape of the time-to-event distribution other than its local linearity.

To begin validating this measure, we apply it to simulated time-to-event data from two processes, one chaotic and one stochastic. Specifically, for a chaotic process, we generate data from the logistic map (18) $x \rightarrow rx(1 - x)$ , with $0 < x < 1$ and parameter $0 < r < 4$ . The logistic map is known to be chaotic for certain parameters $r$. We selected $r = 39$ and measured the distribution of time to event, where an event is considered to occur when the dynamic variable $x$ falls in a certain fixed small region [for us, the interval (0.2…0.21)], and the initial distribution is uniformly sampled from the interval (0…1). For the stochastic process, we sample from a geometric distribution with the same mean, which is the time-to-event distribution of a stochastic Bernoulli process. The resulting time distributions look qualitatively similar for sufficiently long waiting times (see Fig. 1), while the distribution of short waiting times is erratic for the chaotic process. Consequently, detecting chaos in such time-to-event distributions can be expected to be difficult since the chaotic signature is most noticeable only for sufficiently short waiting times.

We find a higher proportion of inversions in the chaotic data as opposed to the stochastic data, and we find statistical significance for rejecting the null hypothesis of a locally linear distribution in the chaotic case ($P < 0.05$) but not in the stochastic one ($P = 0.41$) (see Fig. 2). These results suggest that our inversion measure is capable of discriminating chaos from stochastic behavior using only time-to-event distributions without access to the underlying time series itself.

To further test the ability of the inversion measure to reliably detect a chaotic signature, we applied the same test to time-to-event distributions associated with

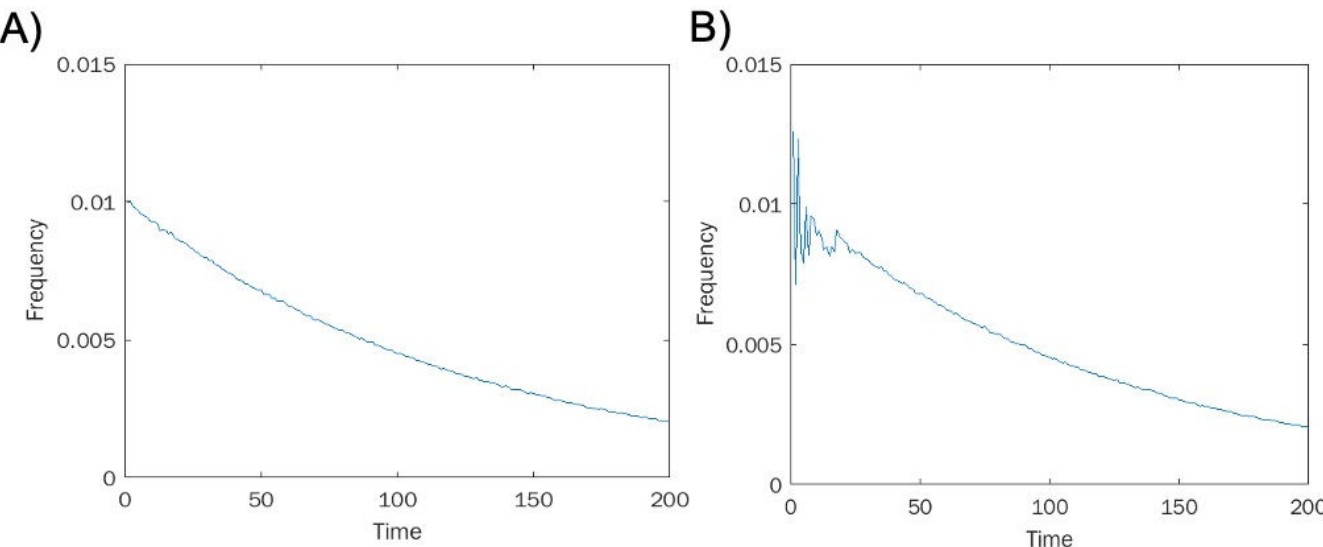

**FIG 1** Time-to-event distributions. (A) Sampling geometric distribution. (B) Sampling from the chaotic quadratic map $x \to rx(1 - x)$, $r = 3.9$ .

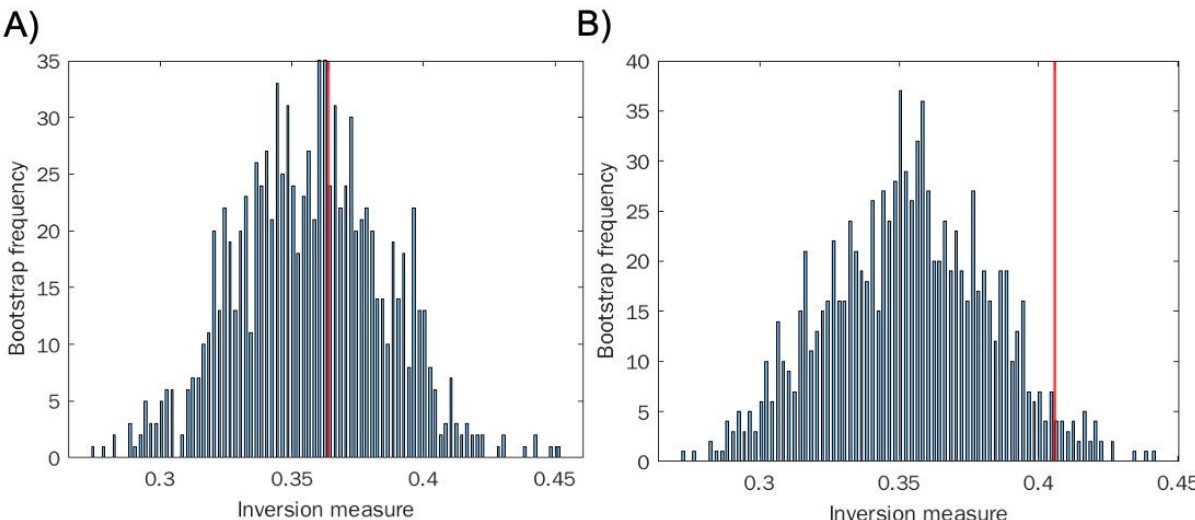

**FIG 2** Inversion method of bootstrap for locally linear approximation of the distributions (histograms). (A) Sampling from a geometric distribution. (B) Sampling from the chaotic quadratic map. Red line indicates the actual inversion measure of the distribution. Cautionary note: the results derived by the inversion method presented in panel A, though in agreement with our hypothesis, are sensitive to parameter choice and can vary from run to run, even with a large sample size. In the text, please see a discussion of the false-positive levels as a function of the histogram's number of bins, for data coming from the Weibull, a generalized exponential distribution.

two other known chaotic systems: the Henon map and the Lorenz system, which are, respectively, two- and three-dimensional. In each of these, we picked a small window in space that intersects the attractor of the system (so it will be repeatedly visited by the system) and defined this window as our "event." The Henon map is given by the dynamics $x_{n+1} = 1 - ax_n^2 + y_n$; $y_{n+1} = bx_n$ . We used the "classical" Henon given by $a = 1.4$, $b = 0.3$, and we defined our event window by $[0.25, 0.35] \times [0.15, 0.25]$. The Lorenz system is a differential equation given by $\dot{x} = \sigma(y - x)$; $\dot{y} = x(\rho - z) - y$; $\dot{z} = xy - \beta z$. We approximated this differential equation by the corresponding difference equation with step size 0.01, using the parameters $\sigma = 10$, $\rho = 28$, and $\beta = 8/3$ (the "classical" Lorenz system used by Lorenz), with the event window given by $[-1.0, 1.0] \times [-1.0, 1.0] \times [13.0, 1.5.0]$. We used 10,000 sample

points for each process. To test the sensitivity of the system to the number of histogram bins, for both the Lorenz and Henon systems, we binned the waiting time data into a range of bins totaling from 50 to 1,000 bins before applying the inversion measure [for both dynamical systems and for all variables, the initial distribution is uniformly sampled from the interval $(0…1)$]. For both systems, the inversion measure again rejects the null hypothesis with statistical significance.

As stochastic control, we sampled data from various Weibull distributions, which is a common family of distributions of waiting times, with a wide variety of shapes. We swept over different choices of parameters for the number of bins for the histogram, and for each one ran the inversion measure multiple times to calculate the percentage of false positives. We found that for a certain range of bin size parameters (around 100–300 bins, given 10,000 sample points), false positives were low for a variety of Weibull distributions (ranging between 3% and 7%). However, if more bins were used, the number of false positives increased significantly. Still, given an appropriate choice for the number of bins, we were broadly able to distinguish between chaotic systems and stochastic ones. Specifically, when using 256 bins on 1,000 runs of 10,000 sample points each for data coming from Weibull distribution, Lorenz, and Henon maps, we obtained a 9.8% false-negative rate for the Lorenz process, an 11% false rate for the Henon process, and a 6.6% false-negative rate for the Weibull distribution. For these tests, we used a kernel-smoothed density (MATLAB function ksdensity with its default settings) as the null hypothesis rather than a local linearization.

We repeated the above experiments with an alternative choice of "terminal event," where the event is given by passing a certain extreme threshold rather than landing inside of a small region, $x(t) < -17$, and $x(t) < -1.1$ for the Lorenz and Henon maps, respectively. In this experiment, we begin by randomly picking the initial conditions for $x$ and $y$ (and $z$ in the case of the Lorenz system) uniformly between 0 and 1 and then iterate the system until it reaches the attractor (to ensure it has entered a chaotic regime). We continue to iterate until $x > 0$, at which point we start measuring the waiting time to the extreme terminal event. Using such a threshold can be seen as a plausible analogy for the transition from "health" to "death," as it involves crossing over into an extreme region of the state space. Using 7,500 data points, we observed a 10.4% false-negative rate for the Lorenz system. Decreasing the number of data points results in higher false-negative rates: 16.2% for 5,000 data points. However, as the number of data points decreases, the false negatives increase drastically (71% for 1,500 data points) due to a lack of statistical power. On the other hand, the Henon map shows lower false-negative rates even for a small sample: 6% false negative for 1,500 data points (which is a comparable number to the sample size of our biological data sets). For comparison, for the Weibull histogram, we observed a 2% false-positive rate for 1,500 data points.

We note that the Lorenz system as shown in Fig. 3 is characterized by erratic flips between two regions characterized by $x > 0$ or $x < 0$. These erratic flips can give additional intuition to the irregular behavior of waiting times for an event if the event is precipitated by such a flip. Still, it is not immediately clear how the erratic nature of these flips translates to irregularities in the histogram of waiting times, shown in Fig. 3

An additional problem associated with the inversion method needs further investigation. The inversion method uses the distribution of time to events as its input rather than a dynamic time series. This fact makes it harder to discriminate between chaotic and stochastic dynamics for certain chaotic systems whose time-to-event stationary distributions are similar to those of stochastic processes, an issue which can be heightened when taking into account the distribution of longer waiting times, for which the chaotic signal tends to fade away.

## Biological data analysis

For the biological model systems described below, time-to-death data are obtained as described in the experimental setup in Materials and Methods, with no preprocessing or change of units. These experiments were conducted using automated *C. elegans* lifespan

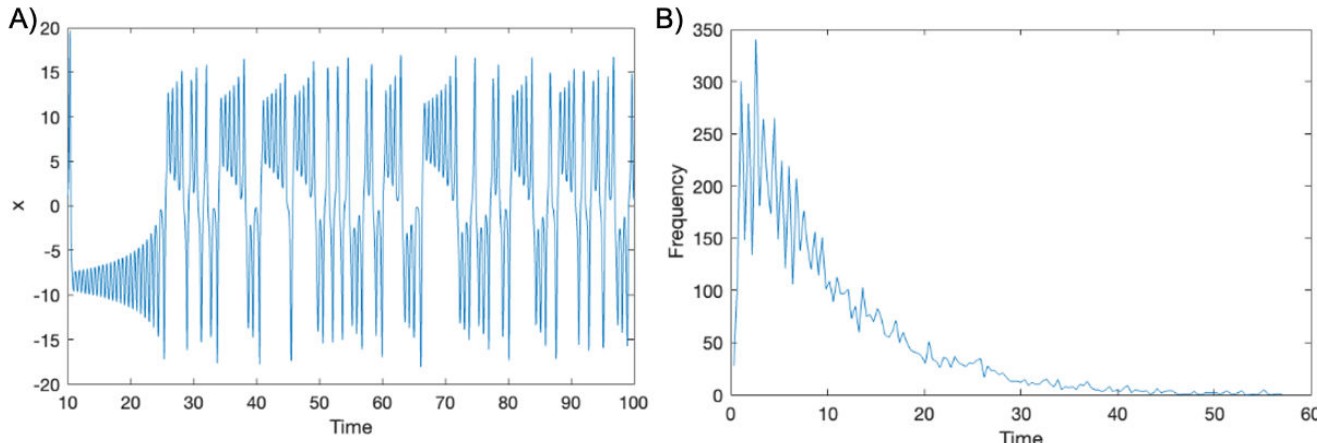

**FIG 3** Characteristic behavior of the Lorenz dynamics. (A) A typical dynamical trajectory of the first variable (*X*) and (B) a histogram of arrival times to an extreme state.

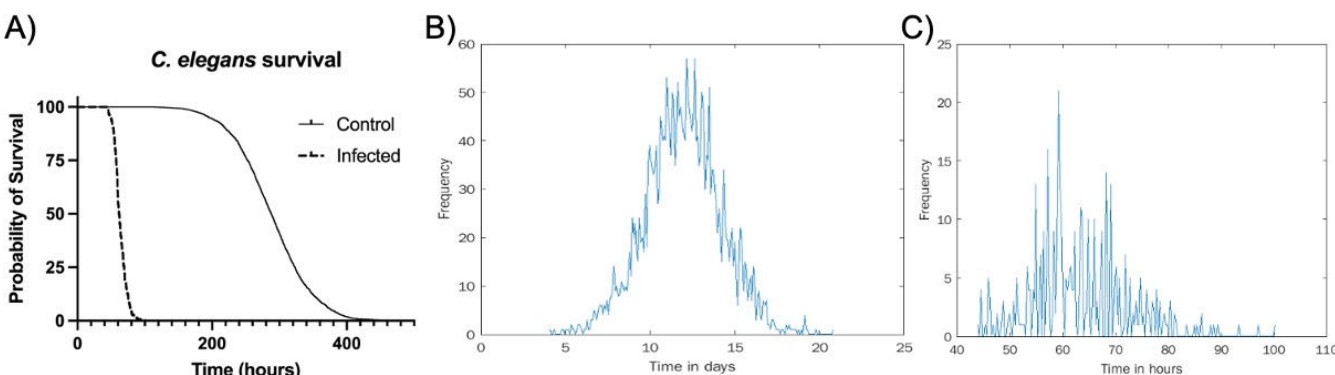

**FIG 4** Survival analysis of non-infected (data from Kulp [21], *n* = 2,992) and infected (data generated in Ausubel lab, *n* = 407) *C. elegans*. (A) Survival over time. Time-to-death histograms for non-infected (B) and infected (C) worms.

machine analysis, which utilizes modified flatbed scanners to capture images of petri plates containing *C. elegans* animals feeding on a non-pathogenic bacterium (*Escherichia coli*) or a pathogenic bacterium (*Pseudomonas aeruginosa*) at set intervals for processing by automated analysis software as described in references 22, 23. This technology enables unbiased, continuous tracking of individual *C. elegans* animals throughout their adult lifespan and has the potential to determine an individual animal's time of death within a 15-min interval or less. Additionally, these experiments can use relatively large populations per condition with each plate containing up to 35 worms. In the case of *D. melanogaster*, visual observations were made to assess the survival of groups of flies (*n* = 20–30 per vial) feeding on normal fly food or a fly food amended with a pathogenic bacterium (*Pseudomonas entomophila*). Dead flies were counted at various time points ranging from every hour up to 48 h.

### Caenorhabditis elegans

We first constructed the time-to-death histograms for both non-infected and infected *C. elegans* data sets to visually determine the differences between them.

As can be observed in Fig. 4, the two histograms differ substantially. While the time-to-death distribution of the non-infected individuals resembles the normal distribution, the infected individuals' histogram is far more irregular. This irregularity may partially result from a smaller sample size of the infected population (Fig. 3). The two

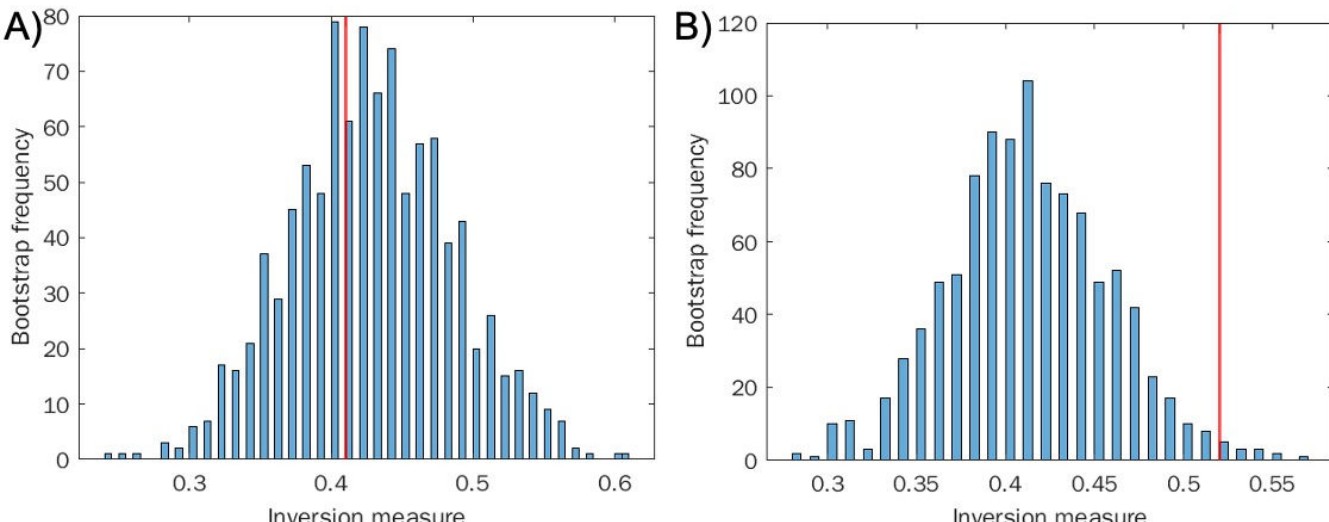

**FIG 5** Inversion method of bootstrap for locally linear approximation of the distributions (histograms). (A) Sampling from non-infected *C. elegans* population distribution. (B) Sampling from infected *C. elegans* population. Red line indicates the actual inversion measure of the distribution.

histograms, however, come from two different underlying biological processes and so cannot be directly compared. The former results from a process of aging-related death, while the latter reflects the pathogenic process of the worm-microbe interaction and the consequent immune system response.

To discriminate whether our system of host-pathogen interactions exhibited chaos, we employed the inversion measure on these time distributions, as described above and in Materials and Methods. We find statistical significance for rejecting the null hypothesis in the infected population ($P < 0.02$), a possible indicator of chaos, and we do not find statistical significance for rejecting the null hypothesis in the uninfected population ($P = 0.64$) (see Fig. 5). Here, we treated the two populations similarly.

This observation, though not definitive due to the preliminary nature of our inversion-based test for chaos, lends credence to the hypothesis that the time-to-death data result from the infected *C. elegans* population is the result of an underlying chaotic dynamical process. As an additional null distribution to compare against, we considered a different smoothing mechanism as opposed to locally linearizing the density: kernel smoothing. We used MATLAB's ksdensity function to smooth our sample distribution and used this smoothed density as the null to generate a bootstrap distribution of inversion measures. We were unable to confidently reject this null hypothesis ($P = 0.11$). However, we believe that for too few data points (see above for the counterargument), the kernel-smoothed density may be too different from the actual density to meaningfully compare against, whereas the locally linearized density closely maintains the local trends. As discussed above, the appropriate number of required data points for the inversion method to be reliable requires further investigation.

### *D. melanogaster* infected with *P. entomophila*

We repeated a similar procedure to analyze infected and non-infected fly data sets (Fig. 6). Here, however, we had to contend with data that were collected manually instead of by machine. While observations were mostly carried out in regular intervals, there was a specific time interval used between data collection points. This means that the number of deaths observed at the subsequent time point needed to be distributed over a wider time interval. To do this, we randomly redistributed the number of deaths observed at wider time intervals into regular subintervals. This introduces an additional element of randomness into the analysis, which we would expect to drown out the chaotic signature. Nevertheless, we still detected statistical significance ($P < 0.05$) in the

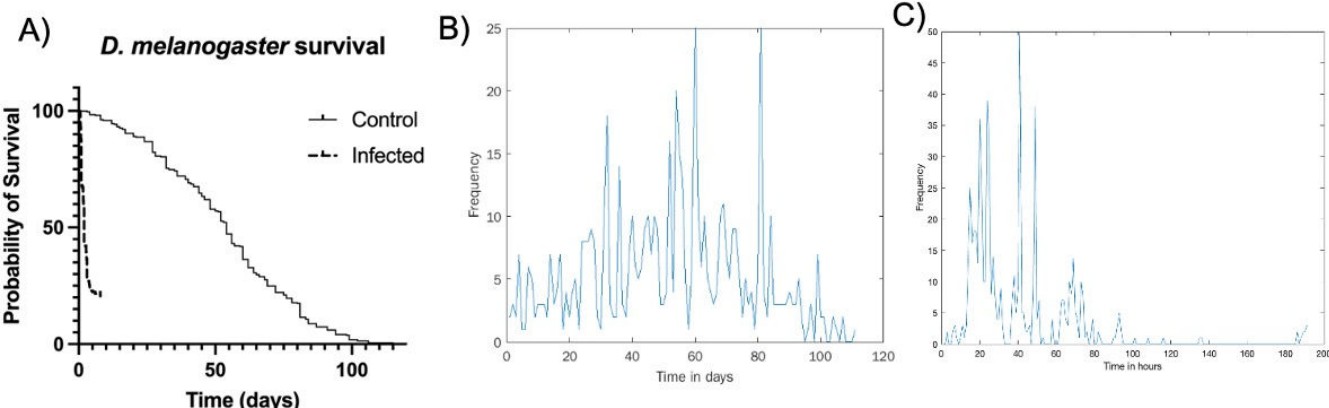

**FIG 6** Survival analysis of non-infected (*n* = 366) and infected (*n* = 673) *D. melanogaster*. (A) Survival over time. Time-to-death histograms for non-infected (B) and infected (C) flies.

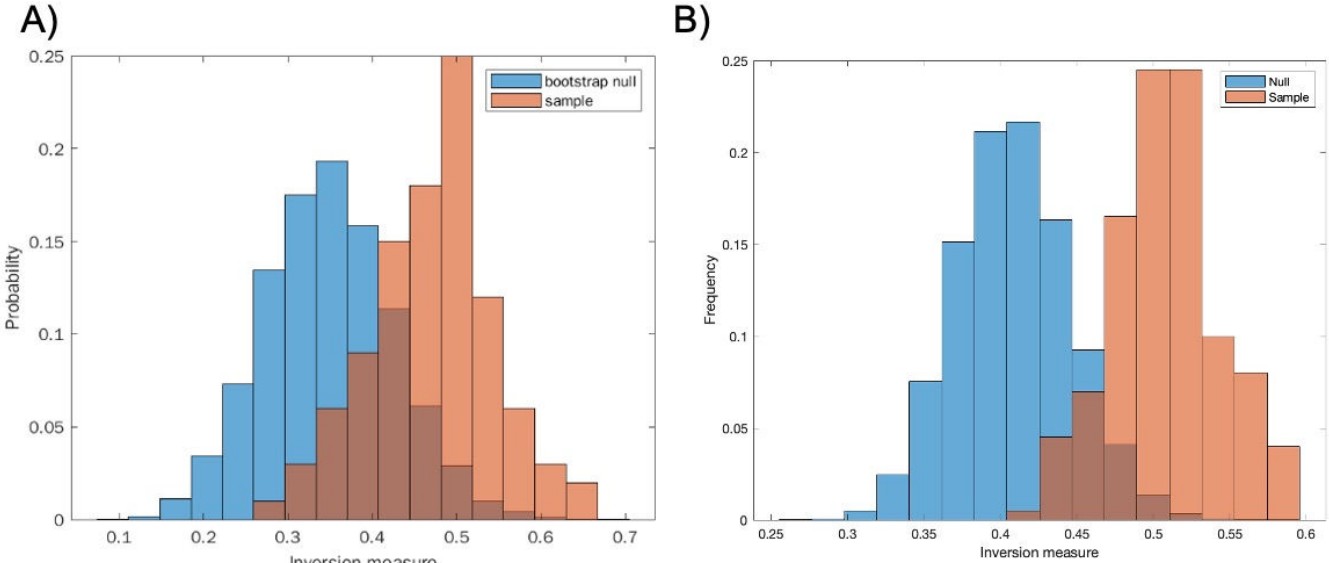

**FIG 7** Comparing the null and sample distributions of the inversion measure. (A) Non-infected. (B) Infected.

inversion measure (Fig. 6). In this case, rather than obtaining a single inversion measure for the histogram, we generated a distribution of inversion measures to account for the randomness of the redistribution process and compared this distribution to the null bootstrap distribution. At the same time, in the uninfected fly data,0 we did not find a statistically significant difference between the bootstrap null and sample distribution (Fig. 7).

## Power analysis

To add context to our results, we conducted a power analysis to determine how many data points can be removed while maintaining statistical significance (see Materials and Methods for more details). We conducted this analysis with the infected worm data and saw that the statistical significance threshold of $P$ = 0.05 was reached around a sample size of 350, compared to our sample size of 407. The same analysis was performed on the data obtained from the infected fly data resulting in a required sample size of a similar order of magnitude (Fig. 8).

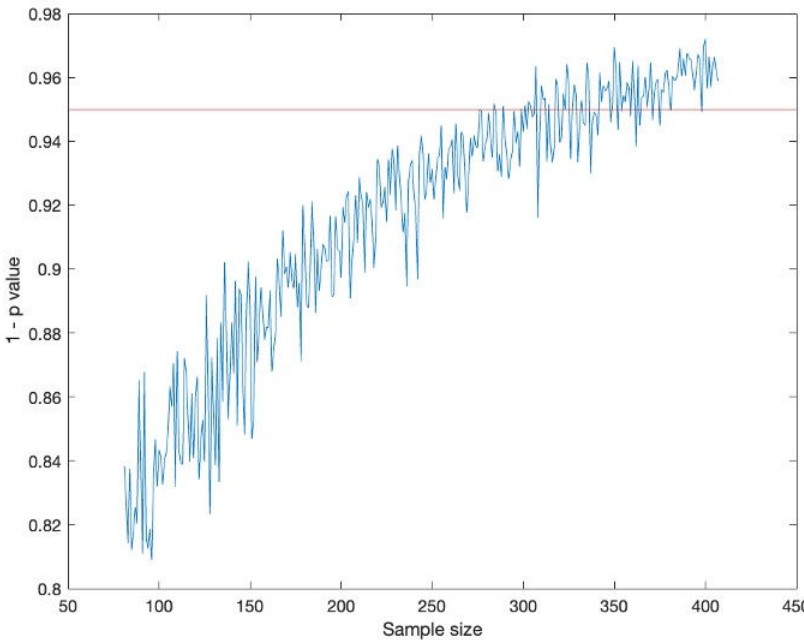

**FIG 8** Power analysis for the *C. elegans* data set.

## DISCUSSION

In this study, we first established that a time-to-death distribution provided adequate information to analyze dynamical signatures in *C. elegans* infected with *P. aeruginosa* and *D. melanogaster* infected with *P. entomophila*, in search for evidence of chaotic behavior. While we found suggestive evidence for chaos in both models with the infection time-to-death series, no such evidence was found in the non-infected populations (normal aging). To our knowledge, this analysis provides the first evidence that host-microbe interactions may be subject to chaotic behavior. Knowledge of whether virulence is deterministic, stochastic, or chaotic, and the types of host-microbe interactions that manifest those properties, is critically important for understanding the fundamental attributes of virulence. Identifying the relevant variables that potentially drive the dynamical behavior in such interactions could have profound implications for public health, vaccine design, and preparedness against emerging pathogens.

The observation of potential chaotic signatures in time to death of a population of infected animals may have important implications for the predictability of microbial pathogenesis. Both models are invertebrates, which lack adaptive immunity. We do not know the generalizability of these observations to other host-microbe systems, but the occurrence of chaotic dynamics in vertebrates would affect our ability to predict future threats. For example, the coronavirus disease 2019 pandemic has highlighted the importance of identifying other potential threats in nature. The challenge of virulence forecasting is highlighted by the fact that there are more than 300,000 viruses among the more than 6,000 known mammalian species (24). Could the subset of mammalian viruses that threaten humans be predicted? Some may argue that the answer to this question must await additional mechanistic information that could be used to model human infections. However, even if we knew all the steps by which a single virus could interact with host cells, the immune response to that virus, the potential for transmissibility, prediction of virulence in a population, etc., it would not be possible if the overall process is stochastic or chaotic. This is because small differences in the host-microbe interaction would have major effects on the outcome of an infection, as manifested by a range of outcomes from asymptomatic infection to pathogenicity, which in the extreme results in the death of the host. An example of this phenomenon is

the unpredictability of infection at the level of the individual, as evident in the current SARS-CoV-2 pandemic where outcomes range from asymptomatic infection to death and are determined by a set of variables that in combination produce uncertain outcomes (3). Chaotic systems are inherently unpredictable although it is possible to generate probabilistic models for short-term prediction of such systems, as is done in applications of stochastic processes and machine learning to weather forecasting. Hence, our finding that some host-microbe interactions may manifest as chaotic dynamical systems implies that absolute predictability with regard to the outcome of these interactions may be impossible if these dynamics are widespread in nature. Nevertheless, knowledge of whether some host-microbe interactions are chaotic could inform efforts to control such diseases including the possibility of exploiting the properties of chaos, itself, to our advantage. For example, once the relevant driving forces have been identified in the analysis of chaotic systems, tools from the field of control theory, in general, and chaos control theory, in particular, can be applied to hypothesize perturbations that could be applied to stabilize a chaotic system (25).

In our prior exploration of chaos with *C. neoformans*-infected *Galleria melonella* (7), we found that the outcome of infection was deterministic and predictable without evidence of chaos. In the course of the work presented here, we have learned that the time-to-death data must be obtained as precisely as possible for each individual host. Likewise, infection time is an important variable. For both worm and fly models, infection occurred shortly after the commencement of the experiment. For our purposes, we are using the definition of infection as the acquisition of the microbe by the host (1). Worms become infected shortly after encountering the *P. aeruginosa* lawn and remain infected for the duration of the experiments, which precludes the possibility of re-infection. Flies are starved prior to being exposed to infected food and thus rapidly acquire infection, which is rapidly damaging to their gut and reduces the likelihood of subsequent infections. Hence, although we cannot control the exact time when the bacteria entered either host, we anticipate that it occurred shortly after inoculation for both systems. For flies, we recognize that after their initial feeding to relieve their hunger, subsequent feeding times can be affected by circadian rhythms, etc., but note that this is physiological for these insects. The alternative of infecting each animal by mechanical injection is not practical since our power calculation shows that hundreds of individuals are needed, which would inevitably require many hours to infect such numbers, and mechanical infection would introduce additional uncertainties including skin damage and likely differences in the puncture site and inoculation depth. In fact, we used this approach in our prior *Galleria melonella* study that failed to identify chaotic signatures (7), probably due to a combination of lack of power and too high inoculum. An additional problem with that prior study is that it evaluated time to death measured at large intervals (e.g., daily), but this is inadequate for detecting dynamical fluctuations on a finer scale, as is the case in our experimental analysis in this paper. Many studies in the field of microbial virulence rely on measuring the percentage of animals that survive daily and round to the nearest 24-h interval irrespective of the fact that each individual animal succumbed at a different time. This is a form of averaging that is done for experimental convenience since it is often not practical to monitor the experiment continuously to record the exact time of death. While exploring methods to study chaos in host-microbe interactions, we found that such averaging could reduce or abolish chaotic signatures. In light of the current analysis, our inability to find chaotic signatures in our prior study may have been a false-negative result stemming from death sampling at large intervals and low power since it analyzed only 240 events (7). Hence, the detection of chaos in host-microbe interactions is highly dependent on experimental design, and the common practice of making measurements at large time intervals produces data that are unsuitable for ascertaining the dynamical nature of the system. Fortunately, with the availability of continuous monitoring devices such as cameras, it is possible to obtain accurate survival data for each individual in the study and thus generate a fine-grained time-to-death histogram that can be analyzed by the methods described here. Though

our current results provide statistically significant conclusions, larger sample sizes and even finer-grained time-to-death measurements could provide even better data sets to solidify our conclusions.

In addition, further validation of our inversion measure and the bootstrap method for obtaining the statistical significance is necessary to solidify our conclusions. In this regard, we identify areas for future investigation, including obtaining a better understanding of how the choice of number (individuals infected) and the size of bins (time intervals) affect the results when a finer time resolution of the data sampling becomes available. In fact, this analysis indicates the need to develop new methods for following the dynamics of host-microbe interactions inside the host since the currently used variables to measure outcome, such as time to death, weight loss, fever, etc., do not have sufficient resolution to provide a conclusive result. In addition, since the inversion measure appears sensitive to parameters such as the number of bins and can result in false positives/negatives in certain parameter settings, further work will be needed to refine the measure and increase its accuracy. Finally, a deeper theoretical study is needed to understand the limits of discerning chaos from time-to-event data, given the nature of ergodic processes.

In recent years, there has been great concern about the reproducibility of biomedical research (26, 27). The detection of chaotic signatures in host-microbe interactions implies that these systems may have inherent challenges to reproducibility. Experimenters studying virulence or the efficacy of a therapeutic intervention in an animal model often choose an inoculum that rapidly kills all members of the infected group, leaving no time for the host-pathogen dynamics to present itself. Such doses can ensure conclusive results with relatively few animals but translate into making the experimental system deterministic. Consequently, experiments that force an outcome are not informative of the dynamical nature of natural host-microbe interactions. However, even when using a large inoculum to ensure mortality, there is considerable variation in such variables as time to death. In this regard, our observation of chaotic signatures suggests that differences in the microbial load administered and/or the site of infection could translate into large differences in effect size, negating the consistency and reproducibility of inter-experimental comparisons.

The fact that variation in experimental conditions is present should not affect the dynamical properties of the underlying dynamical response of the immune system to pathogens. In our simulation, we start with a random initial condition and are nonetheless able to distinguish between chaotic and stochastic dynamical behavior.

In summary, this work should be considered an exploration of the dynamics of host-microbe interactions. While our work suggests that a time-to-death distribution may provide adequate information to analyze for chaotic signatures in host-microbe interactions and that such signatures can be observed in *C. elegans-P. aeruginosa* and *D. melanogaster-P. entomophila* infection systems, we caution that the results were dependent on the parameters chosen for the calculation and that more work is needed before firm conclusions can be made. In fact, we are not certain whether experiments that measure time to death or the proposed inversion method can provide sufficient information to unambiguously determine the dynamical nature of host-microbe interactions and there is a need for exploring other experimental parameters. Nevertheless, the methods described here provide both a starting point for exploring other host-microbe interactions that hopefully will expand our understanding of the fundamental dynamics of underlying infectious diseases and highlight areas of necessary mathematical research to understand how best to study such dynamical processes.

## MATERIALS AND METHODS

### *C. elegans* assays

Wild-type (N2) *C. elegans* were maintained using standard methods. All lifespan and killing assays were performed using the *C. elegans* Lifespan Machine (22, 23). For the

lifespan survival of uninfected worms (N2 worms feeding on *E. coli* OP50), we used the publicly available original data set from references (19, 22). For infection data, assays were performed in the Ausubel lab using N2 worms and *P. aeruginosa* strain PA14 prepared as described in reference 28. Survival data are available on figshare, https://figshare.com/s/0c8fdeccef1098972b73.

### *D. melanogaster* assays

*Pseudomonas entomophila* was streaked from lab glycerol stocks onto LB supplemented with 10% milk and incubated at 29°C overnight (26). Isolated colonies showing lytic activity on milk plates were then inoculated into LB medium and cultured at 29°C for 18 h. Pellets from concentrated cultures (5,000 RPM for 20 min at 4°C, $OD_{600} = 200$) were mixed 1:1 with 2.5% sucrose; for the negative control, 2.5% sucrose was mixed 1:1 with LB. Adult wild-type (Canton-S) flies (4–6 days old, $n = 25$–35 flies/vial) were starved in empty vials for 2 h at 29°C and then transferred to infection vials that consisted of our normal fly food (29) covered with a Whatman filter disc onto which the prepared inoculum or media control was applied (140 µL of the 1:1 bacteria/media:sucrose mix). Flies were kept at 29°C for the duration of the assay but were flipped into new vials without pathogen/infection filters on 24 h post-infection. Pathogen-independent mortality was recorded at 2 h post-infection. Thereafter, the number of dead flies per vial was recorded by visual observation at different (random) time intervals for approximately 7 days after infections. For lifespan survival, cohorts of wild-type (Canton-S) female flies (4–6 days old, $n = 25$–35 flies/vial) were monitored every 1–3 days for survival. Flies were flipped into new media tubes every 3 days. Cohorts were monitored for up to 8 days for infected flies or non-infected flies until all flies in the vial were dead. Survival data are available on figshare, https://figshare.com/s/0c8fdeccef1098972b73.

### The inversion measure on a time distribution

Given a distribution of times, we first convert the distribution into a histogram. In the case when the time points are naturally discrete, such as in our simulated data of waiting times sampled from discrete chaotic or stochastic processes, then we use each discrete time point as a bin. In the case of a continuous process, then we pick a parameter *n* and divide the time distribution into *n* bins.

Given a histogram with n bins, where each bin contains a whole number, we first add a random noise, distributed uniformly at random between 0 and epsilon, where epsilon < 1, to the histogram so as to break ties, ensuring each bin has a unique value while preserving the order of the unequal counts. Then, we subdivide the bins into consecutive nonoverlapping sequences of four bins each, with the remainder discarded. For each such sequence $x_1, x_2, x_3, x_4$, it is a countertrend, or an inversion, if $(x_4 - x_1)$ and $(x_3 - x_2)$ have the same signs (positive or negative). We then calculated the frequency of sequences that are inversions. Since this frequency depends on the randomness of the tie-breaking procedure, we reran this process 1,000 times with different randomizations and took the average.

To obtain a *P*-value for the inversion measure, we used a bootstrap against a null hypothesis that the histogram is smooth. We considered two types of null density: kernel smoothed and locally linear. For the kernel smoothing, we applied MATLAB's built-in ksdensity function to the sample histogram. For the locally linear null density, we locally linearized the sample histogram as follows. For each consecutive sequence of four bins, let $x_1, x_2, x_3, x_4$ be the corresponding sequence of whole numbers. We calculated a line of best fit for $x_i$ as a function of *i* and let this line of best fit be the null density for this sequence. If the line goes below 0, we reset the negative numbers to 0's. We repeated this process for each sequence of four bins, resulting in a piecewise linear null density (which is then normalized to sum to 1). To generate a bootstrap from a null density (either a kernel-smoothed density or a locally linear one), we sampled from it the same number of points as in the original sample and recalculated the inversion

measure. We did this 1,000 times to generate a bootstrapped null distribution of the inversion measure. We then calculated a $t$-statistic and its $P$-value by comparing the sample inversion measure to this null distribution.

For the fly data sets, in which some observations were not made in regular intervals, we redistributed events along gaps as follows. Whenever one or more observations were missed, we randomly uniformly distributed the next-observed number of events along the interval between the given observation and the prior observation. We then formed a histogram from this redistributed data and proceeded as before to calculate the inversion measure. Due to the randomness of redistribution, we carried this process out 1,000 times, thus generating a sample distribution of inversion measures. For each of these 1,000 histograms, we ran the bootstrap process with a sample size of 1,000, thereby generating a total of 1,000,000 inversion measures that make up the null distribution. We used these two distributions to generate a $P$ value by estimating the probability that the sample inversion measure is greater than or equal to the null inversion measure.

For the uninfected flies, we combined two distinct data sets. In this case, we redistributed each data set along gaps as mentioned above and only then combined the two histograms into one and proceeded with the analysis as just described.

## Power analysis

For the power analysis, we varied a parameter $n$ starting from 20% of the full sample size ($n = 82$ for the worm data) to 100% of the full sample size ($n = 407$ for the worm data). For each $n$, we randomly sampled with replacement $n$ data points (i.e., "events") from the given distribution. We used this reduced sample to recalculate a $P$-value for the inversion measure relative to a null distribution, by repeating the same analysis on this reduced sample. We repeated this procedure 100 times for each $n$-value and took the average of the $P$-value.

## ACKNOWLEDGMENTS

N.A.B. is supported in part by R35GM128871. K.M.S. is supported in part by NIH T32GM136577 and F30AG077736. A.C. is supported in part by R01 AI152078, R01 HL059842, and AI052733. A.B. was supported in part by NIH R01-CA164468 and R01-DA033788.

We are indebted to reviewer 2 for helpful comments and criticisms, which led to additional calculations that strengthened the conclusions while also identifying some of the limitations of the approaches taken.

## AUTHOR AFFILIATIONS

[1]Department of Systems and Computational Biology, Albert Einstein College of Medicine, New York City, New York, USA
[2]Department of Biology, Johns Hopkins University, Baltimore, Maryland, USA
[3]Department of Molecular Microbiology and Immunology, Johns Hopkins School of Public Health, Baltimore, Maryland, USA
[4]Department of Molecular Biology, Massachusetts General Hospital, Boston, Massachusetts, USA
[5]Santa Fe Institute, Santa Fe, New Mexico, USA

## AUTHOR ORCIDs

Nichole A. Broderick http://orcid.org/0000-0002-6830-9456
Arturo Casadevall http://orcid.org/0000-0002-9402-9167
Aviv Bergman http://orcid.org/0000-0002-6340-2125

## AUTHOR CONTRIBUTIONS

Yehonatan Sella, Formal analysis, Investigation, Methodology, Writing – original draft | Nichole A. Broderick, Investigation, Supervision, Writing – original draft | Kaitlin M. Stouffer, Formal analysis, Investigation | Deborah L. McEwan, Investigation, Writing – review and editing | Frederick M. Ausubel, Formal analysis, Investigation, Writing – review and editing | Arturo Casadevall, Conceptualization, Investigation, Supervision, Writing – original draft | Aviv Bergman, Conceptualization, Formal analysis, Supervision, Writing – original draft

## ADDITIONAL FILES

The following material is available online.

### Open Peer Review

**PEER REVIEW HISTORY (review-history.pdf).** An accounting of the reviewer comments and feedback.

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
