## [Reviewer comments · mSystems]

Preliminary Evidence for Chaotic signatures in host-microbe interactions

Yehonatan Sella, Nichole Broderick, Kaitlin Stouffer, Deborah McEwan, Frederick Ausubel, Arturo Casadevall, and Aviv Bergman

Corresponding Author(s): Arturo Casadevall, Johns Hopkins Bloomberg School of Public Health

Review Timeline:

Submission Date:	October 16, 2023
Editorial Decision:	November 28, 2023
Revision Received:	December 18, 2023
Accepted:	December 19, 2023

Editor: Seth Bordenstein

Reviewer(s): The reviewers have opted to remain anonymous.

Transaction Report:

DOI: <https://doi.org/10.1128/msystems.01110-23>

Re: mSystems01110-23 (Chaotic signatures in host-microbe interactions)

Dear Dr. Arturo Casadevall:

Thank you and the team for the patience and diligence in revising your work. Below you will find the excellent comments of the two original reviewers, my comments, and instructions from the mSystems editorial office. This article was reviewed by an experimental and theoretical biologist who have improved the manuscript over the course of the revisions and who have different thoughts on the suitability of the work. It's clear to me that this conversation should happen out and in the open for the community to sink into. A new idea often comes with such deliberation...that is the fabric of science, and I for one am a wholehearted supporter of seeing new ideas take shape and form.

I appreciate if you can make the following changes in the revision:

1. Address all of the reviewer comments to ensure the manuscript transparently accommodates each of their points. You can agree or disagree with them, but please do so directly in the manuscript as points of context.
2. Modify the paper title. The title is conclusive about chaotic signatures whereas the abstract and main text are a bit more nuanced. To bring the full paper into alignment, adjust the title to your "suggestions of chaotic signatures" in the abstract or "preliminary evidence of chaotic dynamics" in the importance section.

Revision Guidelines

Sincerely yours,
Seth Bordenstein

Reviewer #1 (Comments for the Author):

The authors measure time-to-death for worms and flies and test if the distribution has signatures of deterministic chaos or stochastic systems. They utilize a newly devised metric to make these inferences.

The experiments in this study control pathogen exposure, but are unable to control exactly when pathogen-induced negative effects begin in each individual. Pathogen exposure is not the same as pathogen infection which is when a microbe escapes normal containment (epithelium, gut chamber, immune suppression) and harms the host. The authors state many factors can influence microbial pathogenesis "dizzying array of variables that include microbe- and host-specific factors, such as inoculum size, route of infection, host state, temperature, associated microbiota, etc."; yet their experiments do not attempt to control for these variables. This variation in the initial host-pathogen interaction occurs at the individual level, but likely has a direct outcome at the system-level which is reported in this work.

Two issues are preventing me from being excited about this work:

1) The current manuscript seeks to test the assumption that host-pathogen interactions are deterministic - "This interpretation assumes that the system is deterministic since it operates on the implicit assumption that if an experimenter could control for all experimental variables the results would be perfectly reproducible from experiment to experiment. Yet, this assumption has never been formally tested..."

The experiments in this paper measure time-to-death in an incredible number of individuals after a highly controlled initial pathogen exposure, without controlling for the initial differences in infection. With this design, I am unsure how the current study can distinguish between the two alternative outcomes - deterministic chaos (i.e., experiments that have small initial differences that affect the outcome) and stochastic effects (randomness). Experimental variation at the post-exposure phase of infection may determine the signature of deterministic chaos that is observed. If breaches at certain gut regions are more/less common, it seems possible that this might alter the distribution in time-to-death to make it not conform to stochastic.

2) The dose of the pathogen was large enough to cause a deterministic fate (host death in all exposed individuals) - thus this system should look like deterministic chaos and not stochastic effects. The very rapid death of hosts (for the most part <100hrs post exposure) is similar to the time-to-death for *Galleria* moths in the citation 7 (Garcia-Solache et al. 2013). In the response the authors state "Please know that the Garcia-Solache reference mentioned by the reviewer [2] is from our group, and we feel that the inability to find chaotic signatures in that study was a false-negative result that reflected both insufficient power and the use of a high inoculum that forced deterministic dynamics on that system." The inoculum amounts in the current experiments might also be too high and force deterministic dynamics on the system.

Furthermore, the time-scale of the deaths due to infection (~4 days) in *Drosophila* makes them susceptible to the influence of daily circadian rhythm-based activities in a way that is not encountered by the deaths in the control population that occurred over the course of months (~100 days). I don't think that deterministic chaos can be measured accurately without taking this into account. This issue is related to Reviewer 2's point #3 - number of bins across histograms "it appears that a very similar number of bins has been used for the non-infected and infected worm data sets, even though the sample sizes differ by more than a factor of 7. That's not what would usually be recommended for constructing a histogram, because it makes things very jagged". The time-scale of an outcome (i.e., days, months) might be the major predictor of observing the outcomes - deterministic chaos or stochastic effects. If the control deaths happened over the course of 4 days (not 100 days), would they also show signs of deterministic chaos?

Reviewer #2 (Comments for the Author):

Please see the uploaded PDF file.

It's a pleasure when a revised manuscript is easy to evaluate, because the authors have fully engaged with the reviewer comments. The other reviewer had concerns very different from mine, which I am not addressing here. Regarding my review of the original submission: the authors have dealt with some of my concerns about validating their "inversion measure" test, and for others they have made it plain to readers that they don't fully know how to deal with the concerns. This paper is tossing a new idea into the ring, and providing enough evidence to suggest that it ought to be taken seriously, while making it clear that there are unresolved issues and so there is further work to be done. Unlike many of the papers that Rogers et al. tossed onto the scrap heap of history (or at least, Rogers et al. proved that they ought to be discarded and forgotten) the revised manuscript makes one makes no extravagant and unjustified claims that will lead others astray.

I'm all for tossing new ideas into the ring if the evidence suggests that they have promise, and I think that the idea in this manuscript now has strong enough supporting evidence. So I will just make two suggestions for further strengthening the paper, and let the Editors and authors wrangle about which of them (if any) ought to be mandatory.

1. The Hénon map, Lorenz system, and Wiebull test cases are done with sample sizes of 10,000. That is a lot more "data" than the empirical examples analyzed later. It would make a much stronger case for your method if the test cases were repeated with a sample size of 1000. To paraphrase an old hit song (and somewhat give away my age), if you don't do it then somebody else will. And if you do it first, you'll have the first chance to interpret the outcome.
2. The time between your original submission and this revision let me come up with one possible answer to the question, why on earth should this possibly be true? For motivation you cite Strogatz's textbook to support the claim that chaotic dynamics are "irregular", which is true but it doesn't provide much rationale for your particular test statistic. And what is the biological mechanism whereby a visit to a particular small region in the middle of a strange attractor could trigger a transition from life to death?

My possible answer is that many chaotic systems will have irregular distributions of the waiting time until some extreme event, or until a flip from one mode of behavior to another, as a function of the initial conditions. Either of those is more plausible as the "trigger" for a transition from health to sickness, than an event defined as visiting some apparently arbitrary small subset of the attractor. Before seeing the revision, I thought that the Lorenz system might be a good example to illustrate that idea. See Figure 1 below. I simulated the Lorenz system from times 0 to 250 to get onto the attractor, and then another 250 times units (the first 50 are shown in the Figure). From the second simulation I chose 1000 random times at which $x(t) < -5$ and determined the waiting time until $x(t)$ became positive. I would guess that this gives a waiting time distribution like the one that you generated, but instead of waiting for some arbitrary region in the attractor, my simulation waits for a qualitative change in system behavior. Similar things could be done for waiting times until z exceeds its 95th percentile.

So finally: my suggestion is to motivate your inversion measure by talking specifically about erratic waiting times until extreme values or mode-flips, and then re-do your examples and test cases with the events being either a value beyond a certain high or low percentile of the state variable's stationary distribution (logistic, Hénon map, or Lorenz z) or a mode-flip

Figure 1: Three panels show the dynamics of the $x, y,$ and z variables of the Lorenz system on its strange attractor. The bottom-right panel shows the distribution of 1000 waiting times until $x(t) > 0$, starting from a randomly chosen time at which $x(t) < -5$.

(Lorenz x or y). Some like my Figure 1 might help readers understand exactly what property of chaotic systems your inversion measure is targeting; or maybe just start your Figure 1 with an additional panel showing the dynamics of the logistic map for $r = 3.9$, so readers can see the erratic distribution of times when extreme values are reached. I can only speak for myself, but thinking about your inversion measure in those terms makes it a lot more plausible to me that it might prove to have some general applicability.

Minor comments

- The first equation in the Lorenz system is $\dot{x} = \sigma(y - x)$, not what you have written in the manuscript. You must have it right in your code – the system in the manuscript blows up.
- For the Hénon map and Lorenz system test cases, please describe how the initial conditions were chosen to generate the time-to-event distribution, as you did for the logistic model.

Point by point response to Editor and Reviewers.

Editor Comments

Thank you and the team for the patience and diligence in revising your work. Below you will find the excellent comments of the two original reviewers, my comments, and instructions from the mSystems editorial office. This article was reviewed by an experimental and theoretical biologist who have improved the manuscript over the course of the revisions and who have different thoughts on the suitability of the work. It's clear to me that this conversation should happen out and in the open for the community to sink into. A new idea often comes with such deliberation...that is the fabric of science, and I for one am a wholehearted supporter of seeing new ideas take shape and form.

We are grateful and appreciative of your editorial approach to this paper. We are breaking new ground as the dynamics of host-pathogen interactions have not been explored in detail and we are being careful with conclusions. We are referring to our paper as 'explorative'. In fact, the work during the revision led us to soften our conclusions from the initial version. This work is highlighting variables that need to be tested.

I appreciate if you can make the following changes in the revision:

1. Address all of the reviewer comments to ensure the manuscript transparently accommodates each of their points. You can agree or disagree with them, but please do so directly in the manuscript as points of context.

Comments addressed below and in the text.

2. Modify the paper title. The title is conclusive about chaotic signatures whereas the abstract and main text are a bit more nuanced. To bring the full paper into alignment, adjust the title to your "suggestions of chaotic signatures" in the abstract or "preliminary evidence of chaotic dynamics" in the importance section.

*Agree – we have added the word preliminary to the title. Title now reads: **Preliminary Evidence for Chaotic signatures in host-microbe interactions***

Reviewer #1 (Comments for the Author):

The authors measure time-to-death for worms and flies and test if the distribution has signatures of deterministic chaos or stochastic systems. They utilize a newly devised metric to make these inferences.

The experiments in this study control pathogen exposure, but are unable to control exactly when

pathogen-induced negative effects begin in each individual. Pathogen exposure is not the same as pathogen infection which is when a microbe escapes normal containment (epithelium, gut chamber, immune suppression) and harms the host.

Response. We respectfully disagree with the terminology used here, which could be contributing to how the different ways that reviewer 1 and the authors of the paper see the results of the study. We agree that exposure is not the same as infection (sitting next to a person with COVID-19 represents an exposure but that does not equal infection, which is acquisition of SARS-CoV-2). However, the definition of infection stated in the criticism is too narrow – for example, Vibrio cholera causes disease by producing a toxin and does not escape containment in the epithelium, gut chamber, etc.

The authors state many factors can influence microbial pathogenesis "dizzying array of variables that include microbe- and host-specific factors, such as inoculum size, route of infection, host state, temperature, associated microbiota, etc."; yet their experiments do not attempt to control for these variables.

We respectfully disagree with the reviewer's statement. At least in the worm model, inoculum size is essentially controlled by having a vast excess of pathogen on the plates compared to the number of worms (many orders of magnitude). Similarly, the route of infection is the same (ingestion), the temperature is the same for all the worms, there is no "microbiota"..... Where there can be variation in the model is how individual worms respond to the massive ingestion of PA14, and this could depend on random variation in the timing or extent of the immune response. Our experiment looks at the perturbation of C. elegans and D. melanogaster hosts by infection and we are studying the dynamics of this system rather than individual responses.

This variation in the initial host-pathogen interaction occurs at the individual level, but likely has a direct outcome at the system-level which is reported in this work.

We agree that the individual variations would probably affect the overall outcome of this system but this criticism does not negate that the overall systems manifest dynamics that can be studied in time series. In fact, as we state above, the initial differences in infection are probably negligible. What is different is how individual worms respond to the infection.

Two issues are preventing me from being excited about this work:

1) The current manuscript seeks to test the assumption that host-pathogen interactions are deterministic - "This interpretation assumes that the system is deterministic since it operates on the implicit assumption that if an experimenter could control for all experimental variables the results would be perfectly reproducible from experiment to experiment. Yet, this assumption has never been formally tested..."

We agree that this assumption has never been tested because it is an impossible experiment with current technology. However, the assumption can host-pathogen interactions are deterministic be made as a thought experiment (or thought hypothesis) of the type that are common in physics and have driven some of the most fundamental questions in quantum mechanism (think about Schrodinger's cat or Einstein-Podolsky-Rosen hidden variable critique). Biology has less experience with thought experiments because it has not frequently had to deal with problems that it cannot approach experimentally. In the revision we have clarified in the Introduction that this is a 'thought experiment' or 'thought hypothesis'

that can be postulated event though it is currently beyond our experimental capabilities.

The experiments in this paper measure time-to-death in an incredible number of individuals after a highly controlled initial pathogen exposure, without controlling for the initial differences in infection. With this design, I am unsure how the current study can distinguish between the two alternative outcomes - deterministic chaos (i.e., experiments that have small initial differences that affect the outcome) and stochastic effects (randomness). Experimental variation at the post-exposure phase of infection may determine the signature of deterministic chaos that is observed. If breaches at certain gut regions are more/less common, it seems possible that this might alter the distribution in time-to-death to make it not conform to stochastic.

Certainly, the reviewer may have a point regarding the lack of complete control over the experimental conditions. Nonetheless, the hallmark of chaotic systems lies in their susceptibility to initial conditions, where minor variances can lead to vastly different outcomes, and the relationship between initial and final states weakens exponentially over time. Conversely, in stochastic systems, the noise seen in the outcomes often mirrors the noise present in the initial conditions. It should be noted that the underlying system governing the dynamical properties remains the same independent of experimental variation. The fact that variation in experimental condition is present should not affect the dynamical properties of the underlying dynamical response of the immune system for pathogens. In our simulation we starts with random initial condition. nonetheless we are able to distinguish between chaoting and stochastic dynamical behavior.

2) The dose of the pathogen was large enough to cause a deterministic fate (host death in all exposed individuals) - thus this system should look like deterministic chaos and not stochastic effects. The very rapid death of hosts (for the most part <100hrs post exposure) is similar to the time-to-death for Galleria moths in the citation 7 (Garcia-Solache et al. 2013). In the response the authors state "Please know that the Garcia-Solache reference mentioned by the reviewer [2] is from our group, and we feel that the inability to find chaotic signatures in that study was a false-negative result that reflected both insufficient power and the use of a high inoculum that forced deterministic dynamics on that system." The inoculum amounts in the current experiments might also be too high and force deterministic dynamics on the system.

We agree that the inoculum used was deadly and could have reduced the chaotic signals by driving a high mortality, but we still saw evidence of chaos. We have taken pains to admit that this is an exploration of the dynamics of the system and future work needs to examine the effect of inoculum dose on system dynamics. However, we have to start somewhere and this work is contributing to testing the feasibility of using time to death in a population to study dynamics of host-microbe interactions. In the revised manuscript we are very careful in expressing nuance when discussing the results.

Furthermore, the time-scale of the deaths due to infection (~4 days) in Drosophila makes them susceptible to the influence of daily circadian rhythm-based activities in a way that is not encountered by the deaths in the control population that occurred over the course of months (~100 days).

We don't understand this criticism since both control and infected groups will be influenced by circadian rhythms, which are part of the normal biology of this species.

I don't think that deterministic chaos can be measured accurately without taking this into account. This issue is related to Reviewer 2's point #3 - number of bins across histograms "it appears that a very similar number of bins has been used for the non-infected and infected worm data sets, even though the sample sizes differ by more than a factor of 7. That's not what would usually be recommended for constructing a histogram, because it makes things very jagged". The time-scale of an outcome (i.e., days, months) might be the major predictor of observing the outcomes - deterministic chaos or stochastic effects. If the control deaths happened over the course of 4 days (not 100 days), would they also show signs of deterministic chaos?

Please see response to Reviewer 2, where we have addressed this issue.

Reviewer 2.

To address the two major points of reviewer 2 we added the following in the the main part of the paper:

'We repeated the above experiments with an alternative choice of "terminal event", where the event is given by passing a certain extreme threshold rather than landing inside of a small region, $x(t) < -17$, and $x(t) < -1.1$ for the Lorenz and Henon maps respectively. In this experiment we begin by randomly picking the initial conditions for x and y (and z in the case of the Lorenz system) uniformly between 0 and 1, then iterating the system for a long time until it reaches the attractor (to ensure it has entered a chaotic regime), then continuing to iterate until $x > 0$, at which point we start measuring the waiting time to the extreme terminal event. Using such a threshold can be seen as a plausible analogy for the transition from "health" to "death", as it involves crossing-over into an extreme region of the state space. Using 7500 data points, we observe a 10.4% false negative rate for the Lorenz system. Decreasing the number of data points results in higher false negative rates: 16.2% for 5000 data points, However, as the number of data points decreases, the false negatives increase drastically (71% for 1500 data points) due to lack of statistical power. On the other hand, the Henon map shows lower false negative rates even for a small sample : 6% false negative for 1500 data points (which is a comparable number to the sample size of our biological datasets). For comparison, for the Weibull histogram, we observed 2.8% false positive rate for 1500 data points.

We note that the Lorenz system as shown in Figure 3 is characterized by erratic flips between two regions characterized by $x > 0$ or $x < 0$. These erratic flips can give an additional intuition to the irregular behavior of waiting times to event, if the event is precipitated by such a flip. Still, it is not immediately clear how the erratic nature of these flips translates to irregularities in the histogram of waiting times, shown in Figure 3

Figure 3: Characteristic behavior of the Lorenz dynamics: on the left we see a typical dynamical trajectory of the first variable (x), while on the right is a histogram of arrival time to an extreme state (see text above).

Also we have corrected the first equation of the Lorenz dynamics and describe the initial conditions for both the Lorenz and Henon dynamical systems.

Re: mSystems01110-23R1 (Preliminary Evidence for Chaotic signatures in host-microbe interactions)

Dear Dr. Arturo Casadevall & Colleagues:

Happy Holidays and thank you again for your patience and deliberation. I am very pleased to report that I reviewed the response/edits, and your manuscript has now been editorially accepted. I am forwarding it to the ASM production staff for publication. Congratulations. Your paper will first be checked to make sure all elements meet the technical requirements. ASM staff will contact you if anything needs to be revised before copyediting and production can begin. Otherwise, you will be notified when your proofs are ready to be viewed.

Featured Image Submissions: If you would like to submit a potential Featured Image, please email a file and a short legend to mSystems@asmusa.org. Please note that we can only consider images that (i) the authors created or own and (ii) have not been previously published. By submitting, you agree that the image can be used under the same terms as the published article. File requirements: square dimensions (4" x 4"), 300 dpi resolution, RGB colorspace, TIF file format.

Sincerely,
Seth Bordenstein
Editor
mSystems